# The Prognostic Role of Preoperative Hematological and Inflammatory Indices in Canine Appendicular Osteosarcoma

**DOI:** 10.3390/vetsci10080495

**Published:** 2023-08-01

**Authors:** Konstantinos Rigas, Jean-Benoit Tanis, Emanuela Morello, Gerry Polton, Laura Marconato, Marlon Carroll, EstelLa Ciriano Cerda, Sofia Ramos, Charlotte Baker, Riccardo Finotello

**Affiliations:** 1Department of Small Animal Clinical Sciences, Institute of Veterinary Science, University of Liverpool, Neston CH64 7TE, UK; kostas.rigas@southfields.co.uk (K.R.); riccardofi@libero.it (R.F.); 2Southfields Veterinary Specialists, Basildon SS14 3AP, UK; 3Department of Molecular and Clinical Cancer Medicine, Institute of Systems, Molecular and Integrative Biology, University of Liverpool, Liverpool L7 8TX, UK; 4Department of Veterinary Sciences, University of Turin, 10095 Grugliasco, Italy; 5North Downs Specialist Referrals, Bletchingley RH1 4QP, UK; 6Department of Veterinary Medical Sciences, University of Bologna, 40126 Bologna, Italy; 7Northwest Veterinary Specialists, Runcorn WA7 3FW, UK; 8Department of Veterinary Medicine, University of Bari, 70010 Valenzano, Italy

**Keywords:** canine osteosarcoma, cut-off, prognosis, sighthound

## Abstract

**Simple Summary:**

Osteosarcoma (OSA) is an aggressive bone tumor both in human and canine patients, commonly treated with surgery and follow-up intravenous chemotherapy. Prognosis is guarded to poor; however, this can be influenced by a variety of factors such as tumor location and serum alkaline phosphatase, among others. Hematological indices have proven to play a prognostic role in humans, but data are limited in dogs. The aim of this retrospective study was to investigate the prognostic significance of pre-operative hematological/inflammatory indices, together with other prognostic factors, in a cohort of client-owned dogs with appendicular OSA treated with limb amputation and injectable chemotherapy. As sighthounds have a predisposition to OSA but present different hematological reference values compared to other breeds, these were also evaluated separately. Fifty-nine dogs were included, and 13 were sighthounds. Data analysis suggests that pre-treatment absolute neutrophil count may play a prognostic role in canine OSA treated with amputation and adjuvant carboplatin. Our study also suggests that hematological markers identified in the whole population may not be applicable to sighthounds. These pre-treatment indices, which could prove very helpful as they are readily available, should be confirmed in large prospective studies considering breed specificities.

**Abstract:**

Hematological indices play a prognostic role in human osteosarcoma (OSA), but data are limited in dogs. The aim of this retrospective multicentric cohort study was to investigate the prognostic significance of pre-operative hematological/inflammatory indices in a cohort of client-owned dogs with appendicular OSA receiving standardized treatment. Cut-offs associated with progression-free survival (PFS) for pre-operative hematological values/ratios were established using the minimal *p*-value approach. Historical prognostic factors were also assessed. Statistical analyses were performed for the whole population and after the exclusion of sighthounds. Fifty-nine dogs were included (13 were sighthounds). Multivariable analysis revealed that a low neutrophil count (<4.37 × 10^9^/L, HR0.28, CI 95% 0.13–0.61, *p* = 0.001), a high red blood cell count (≥7.91, HR3.5, CI 95% 1.56–7.9, *p* = 0.002), and a proximal humerus location (HR3.0, CI 95% 1.48–6.1, *p* = 0.002) were associated with shorter PFS. In the sighthound-only population, only OSA location was significantly associated with PFS in univariable analysis. When sighthounds were excluded, a low neutrophil count, a low monocyte count, and a proximal humerus location were associated with shorter PFS, in multivariable analysis. Neutrophil count and possibly monocyte and red blood cell counts can be useful prognostic markers in canine OSA treated with amputation and adjuvant carboplatin. However, not all indices are appropriate in sighthounds.

## 1. Introduction

Osteosarcoma (OSA) is the most common primary malignant bone neoplasm in dogs, with a median age of seven years but a small peak incidence of between 18 and 24 months [1]. The appendicular skeleton is the most common site affected [1]. Definitive-intent treatment for OSAs includes the combination of surgical intervention and adjuvant chemotherapy, consisting of the intravenous administration of platinum agents or anthracyclines with comparable results [2,3,4].

Despite the variety of treatments investigated and refined over the years, the prognosis is still considered guarded to poor, and most dogs will die due to disease progression. Various negative prognostic factors have been identified, with the most consistent being increased serum alkaline phosphatase (ALP) at the time of diagnosis and proximal humerus location [5]. However, given the importance of the immune response in anti-tumor immunity and tumor-associated inflammation, some attention has also been directed toward hematological/inflammatory indices, and their potential prognostic role [6,7]. One study has in fact associated the increase in circulating monocytes and lymphocytes with a negative outcome [6], while another has observed a negative association between the CD8^+^/Tregs ratio and survival [7]. Many other hematological/inflammatory indices have also been useful prognosticators in other canine tumors such as lymphoma, melanoma, mast cell tumor, and soft tissue sarcomas [8,9,10,11,12,13].

However, data remain fragmented, still limiting the utility of these parameters in veterinary clinical practice. Moreover, previous canine OSA studies have not analyzed sighthound separately, potentially affecting or limiting the results [14]. It should be noted that this group of breeds, which is predisposed to OSA, is characterized by hematological values that are physiologically different from other breeds and not necessarily comparable [14].

Conversely, more extensive knowledge has been acquired in human OSA research, where several reports have strengthened the prognostic role of hematological/inflammatory indices. Indeed, an increased absolute lymphocyte count, a high neutrophil-to-lymphocyte ratio, a low lymphocyte-to-monocyte ratio, a high platelet count, or a high systemic inflammatory index (SII) have been correlated with poor survival [15,16,17,18,19,20,21,22].

The aim of our study was to investigate several hematological/inflammatory indices in a homogeneous population of canine appendicular OSAs treated with surgery and a standardized chemotherapy protocol. The secondary objective of the study was to assess whether hematological prognostic markers, identified in the general population of dogs, could be also applicable to sighthounds.

## 2. Materials and Methods

Medical records of the Small Animal Teaching Hospital of Liverpool University (United Kingdom), Veterinary Teaching Hospital of Turin University (Italy), Veterinary Teaching Hospital “Giuseppe Gentile” of Bologna University (Italy), North Downs Specialist Referrals (United Kingdom), Southfields Veterinary Specialists (United Kingdom), and North West Surgeons (United Kingdom) were retrospectively reviewed to identify client-owned dogs with newly diagnosed appendicular OSA between April 2010 and April 2021.

Inclusion criteria comprised (1) dogs with newly diagnosed, histologically confirmed appendicular OSA, (2) no documented distant metastases (including lungs), (3) treated with amputation and adjuvant carboplatin as a single agent (a minimum of one dose was acceptable for inclusion), (4) with the availability of pre-operative hematology and (5) adequate follow-up. Hematology was considered “pre-operative” if it was performed ≤21 days before surgery. Adequate follow-up was defined as ≥180 days follow-up post-operatively or as death/progression documented within this timeframe. Although only adjuvant carboplatin was allowed as a first-line treatment, there were no exclusion criteria applied to the treatments used at the time of progression. Dogs that had received glucocorticoids and/or neoadjuvant oncological therapies (i.e., radiotherapy, immunotherapy, and chemotherapy) were excluded from the study.

The following data were retrieved from the medical records: Signalment (sex, neutering status, age, and body weight), the presence of lameness, tumor location, date of surgery, date and imaging modalities used for initial staging work-up and restaging (when applicable), date of first carboplatin treatment, dose and number of carboplatin used, date and type of progression, and date and cause of death. Adverse events (AE) leading to changes in chemotherapy administration doses and/or schedule were collected. Chemotherapy AE was classified based on the Veterinary Cooperative Oncology Group—Common Terminology Criteria for Adverse Events (VCOG-CTCAE) following carboplatin [23].

Pre-operative hematological parameters were retrieved from the hematology report. Hematological analyses were performed by a validated commercial veterinary laboratory or a university veterinary laboratory. In all cases, whole-blood samples were processed in a timely manner in EDTA tubes by a validated automated analyzer, and a manual differential count was performed. The platelet count was manually assessed. Table 1 shows the reference intervals used based on an internationally accepted reference range [24]. Preoperative ALP, performed ≤21 days prior to amputation, was also retrieved from the biochemistry report.

Diagnostic imaging techniques, surgical limb amputations, and histological assessments were performed by board-certified or board-eligible veterinarians with competences in each field, or by a resident in training under the guidance of a board-certified veterinary specialist.

The associations between outcome and hematocrit, plateletcrit, mean platelet volume (MPV), red blood cell (RBC) count, absolute lymphocyte, monocyte, neutrophil, and platelet count, and the following hematology ratios were assessed: Neutrophil/lymphocyte (NLR), lymphocyte/monocyte (LMR), platelet/lymphocyte (PLR), platelet/neutrophil (PNR), platelet/monocyte (PMR), neutrophil/monocyte (NMR), and systemic inflammatory index [SII; (platelets × neutrophils)/lymphocytes]. Additionally, the associations between outcome and age, sex, body weight, ALP, time from amputation to chemotherapy initiation, and tumor location were assessed due to their historical prognostic role and/or to limit their confounding effect in prognosis prediction [5,25]. The association between the imaging modality used for staging work-up and outcomes was also assessed. Additionally, as sighthounds can have different normal ranges for hematological values, statistical analysis was repeated excluding them from the population [14].

Due to the subjectivity of the decision to perform euthanasia, the outcome of interest was progression-free survival (PFS), whereas overall survival time (OST) was a secondary endpoint. Progression-free survival was calculated as the time from amputation to the time of documented disease progression or death. Both confirmed and suspected progressions were considered events for PFS. Progression was either confirmed by cytology/histopathology or suspected based on imaging findings and, in this case, cytological/histological confirmation was not required. Overall survival time was calculated as the time from amputation to the date of death by any cause. Death was attributed to OSA when progression was documented as a cause of death/euthanasia (i.e., confirmed OSA-related death) or if clinical signs were suggestive of OSA progression and no other cause could be identified (i.e., suspected OSA-related death). For both outcome measures, patients were censored at the date of last follow-up if still alive and/or disease-free.

The association between continuous variables and outcome was analyzed via a biomarker cut-optimization web application tool *Cutoff Finder*, for both PFS and OST [26,27]. The “Survival” method of this webtool was used. This method fits Cox proportional hazard models to the dichotomized variable and the survival variable. Multiple cut-offs are then iteratively assessed, and the optimal cut-off is defined as the point associated with the lowest *p*-value on the log-rank test. Survival analysis is executed using the functions coxph and survfit from the R package survival [26,27,28,29]. For categorical variables, the log rank test was used to assess the association with the outcome. Given the risk of type I error related to the use of the “minimal *p*-value approach”, only cut-off points associated with *p*-value < 0.05 were included in the multivariable analysis. For the other categorical variables (such as tumor location, sex, or staging modality), each variable associated with the outcome with a *p*-value < 0.1 was included in multivariable analysis. Cox proportional hazard regression analysis was used for multivariable analysis. Hazard ratios (HRs) including 95% confidence intervals (95% CI) were calculated for both progression/relapse and death. HR is equivalent to the odds that a dog in the group with the higher hazard reaches the endpoint first, i.e., progression/relapse for PFS analysis and death for OST analysis. A *p*-value < 0.05 was considered significant. All analyses were performed using R x64bit and R studio [29]. The package ggplot2 and GraphPad Prism version 9.3.1 (GraphPad Software, San Diego, CA, USA) were used for graphical representation [30].

## 3. Results

### 3.1. Study Population

Fifty-nine dogs were included. There were 13 sighthounds: Greyhounds (9), lurchers (2), and deerhounds (2). Among the remaining 46 dogs, the most represented were Rottweilers (9), crossbreeds (9), German shepherds (7), and Labrador retrievers (6). The median age at the time of OSA diagnosis was 8.8 years (range 1.5–13.5) and the median body weight was 33.0 kg (range 6.2–54 kg). There were 34 females (26 neutered, 8 intact) and 25 male dogs (11 castrated, 14 intact). Lameness was reported in 58 dogs, and the median duration of lameness prior to presentation was 19 days (range of 1–143). The OSA was located in the femur in 15 cases, the tibia in 15 cases, the proximal humerus in 13 cases, the radius in 12 cases, the ulna in 2 cases, and the scapula and mid humerus in 1 case each. Demographics and presenting signs of the whole population are summarized in Table 2. A detailed description of each case is documented in Appendix A.

### 3.2. Hematology/Biochemistry

The median time between hematology and amputation was 5 days (range 0–21). The median neutrophil count was 5.45 × 10^9^/L (range 2.3–10.5). Neutrophil count was normal in 56 (94.9%) cases, and 3 (5.1%) dogs were mildly neutropenic (range for neutropenic dogs: 2.3–2.9 × 10^9^/L). Two of these three neutropenic dogs were greyhounds. The median monocyte count was 0.44 × 10^9^/L (range 0.06–1.5): A low monocyte count was noted in 2 dogs (3.4%–0.06 and 0.15 × 10^9^/L), a high monocyte count was noted in 1 dog (1.7%–1.5 × 10^9^/L), and it was normal for 56 (94.9%) dogs. The median platelet count was 295 × 10^9^/L (range 83–622). The platelet count was normal in 40 (67.8%) cases, mild to moderate thrombocytopenia was documented in 12 (20.3%) cases (range 83–185 × 10^9^/L), and mild thrombocytosis was reported in 7 cases (11.9%, range 517–622 × 10^9^/L). One Cavalier King Charles spaniel was included in the cohort and the platelet count was normal (468 × 10^9^/L). The median hematocrit was 48% (range 30.4–65.5): Anemia was documented in five cases (8.5%, range for anemic dogs 30.4–35%). The distribution of the hematological parameters and associated ratios in the whole population are summarized in Appendix A. ALP was available in 55 dogs (Appendix A) and its median was 89 U/L (range 11–675.4 IU/L). ALP was normal in 39 (66.1%) cases and increased in 16 (27.1%, range 147–675.4 IU/L).

### 3.3. Staging

Thoracic imaging was performed in all cases with computed tomography (CT; *n* = 47, 79.7%) or radiographs (*n* = 12, 20.3%). Abdominal imaging was performed in 38 (64.4%) cases by means of CT (*n* = 34, 89.5%), ultrasound (*n* = 3, 7.8%), or radiography (*n* = 1, 1.7%). Imaging of the affected limb was performed in 34 (57.6%) cases with CT (*n* = 28, 82.4%), radiography (*n* = 5, 14.7%), and magnetic resonance imaging (*n* = 1, 2.9%). No metastases or significant comorbidities were identified in any of the dogs (Table 2 and Appendix A). While lymph node metastases were not reported in any dogs, this was not consistently assessed in this study.

### 3.4. Chemotherapy and AE

The median time from amputation to chemotherapy initiation was 19 days (range 3–56 days), and only one dog had chemotherapy that started within five days of amputation. The median number of doses administered was 4 (range 1–6) (Appendix A). Among the 16 dogs who received fewer than four carboplatin treatments, disease progression was the cause of treatment discontinuation in 13 dogs and the reason was unknown in 3 cases.

The median starting carboplatin dose was 300 mg/m^2^ (range 250–300 mg/m^2^); in five cases, the chemotherapy dose was escalated: From 250 to 300 mg/m^2^ in four dogs and from 250 to 279 mg/m^2^ in one dog (Appendix A). Among dogs that received >1 carboplatin, all but two received carboplatin every three weeks. Protocol alteration was performed in nine dogs. Dose reduction alone was performed in seven dogs and the reason was known for six cases: VCOG grade II lethargy in two dogs, VCOG grade I neutropenia in one dog (greyhound), VCOG grade II neutropenia in one dog, VCOG grade II lethargy/grade III diarrhea in one dog, and VCOG grade II lethargy/grade II diarrhea in one dog. Dose reduction and chemotherapy schedule deviation (from three to four weeks) were performed in two dogs: One due to VCOG grade IV neutropenia/grade II diarrhea and the other one due to VCOG grade II neutropenia. The chemotherapy dose was not escalated subsequently in any of these dogs.

### 3.5. Outcome

Restaging was performed in 55 (93.2%) cases. Imaging modalities used at the time of restaging included thoracic radiography (*n* = 32) and thoracic CT (*n* = 23), abdominal CT (*n* = 4), and abdominal ultrasound (*n* = 6). Restaging was only performed at the time of progression in 33 dogs (55.9%). For the other dogs, a median of one (range 1–5) repeat staging workup was performed before the documented progression, death, or until the date of last contact. The date of the last restaging before progression was known for 26 dogs. The median time between the last restaging and the documented progression was 112 days (range 30–319) for these cases.

Progression was suspected upon imaging in 42 (71.2%) dogs at the following sites: Pulmonary parenchyma (*n* = 27, 64.3%), appendicular skeleton (*n* = 8, 19%), spinal vertebrae (*n* = 7, 11.9%), liver (*n* = 3, 7.1%), ribs (*n* = 3, 7.1%), spleen, locoregional lymph nodes, kidney, and subcutis in one case each (2.4%). Metastatic disease was seen in more than one site in eight cases (18.2%). In 17 (28.8%) cases, there was no documentation of progressive disease on imaging +/− cytology/histopathology (Appendix A). Among those, death was ultimately assumed to be attributed to OSA progression in 11 cases, 2 died from unrelated causes, and 4 dogs were lost to follow-up or still alive and disease-free at the time of last contact at 222, 283, 581, and 1493 days after amputation.

The overall median PFS for the whole population was 142 days (95% CI 121–184) and 139 days (range 34–1476) for dogs with documented progression or confirmed death (*n* = 55). Upon progression, rescue treatment was given to 10 cases including doxorubicin (*n* = 1), metronomic cyclophosphamide (*n* = 4), other types of metronomic chemotherapy (*n* = 1: Combination of thalidomide, cyclophosphamide, methotrexate, and non-steroidal inflammatory drug), toceranib phosphate (*n* = 3), or prednisolone (*n* = 1).

### 3.6. Overall Survival Time

The median OST was 201 days (95% CI 180–281). Among the whole population, 48 dogs were dead with a median OST of 181 days (range 36–1476) and 11 were still alive or lost to follow-up after a median of 283 days (range 88–1493). Death was attributed to OSA in 44 cases, while in 4 cases, death was unrelated or the role of OSA could not be confirmed (degenerative disk disease in one case, chronic kidney disease deterioration in two cases, and acute pancreatitis in one case).

### 3.7. Analysis of Possible Prognostic Factors

#### 3.7.1. Whole Population

The results of univariable and multivariable analyses for the association between PFS and OST and the possible prognostic factors and the respective best cut-off values are summarized in Table 3 and Appendix A. In the univariable analysis, a significant association between PFS and the following predictive variables, dichotomized into two groups based on the minimal *p*-value approach, was identified: Absolute count of neutrophils, monocytes, platelets and RBC, hematocrit, plateletcrit, and NLR and NMR. Among these cut-off values, the stability and significance of the possible dichotomization varied across the cut-off values evaluated.

Indeed, the graphs of the HR for each cut-off tested (Figure 1) showed that, among the cut-offs tested for significance, 6 to 15 (18.1% to 44.1%) cut-off values were significantly associated with PFS for the neutrophil count (Figure 1A), platelet count (Figure 1C), and hematocrit (Figure 1F), whereas 1 one to 4 cut-off values (2.6% to 10.8%) were significantly associated with PFS for the monocyte count (Figure 1B), the plateletcrit (Figure 1D), the RBC count (Figure 1E), the NLR (Figure 1G), and the NMR (Figure 1H).

These graphs allow us to assess the stability and significance of the dichotomizations evaluated, thereby helping to comment on the relevance of the optimal cut-off identified. For example, for the RBC count, only 10.8% of cut-offs tested were significantly associated with PFS. This suggests that the dichotomization based on RBC count is rarely relevant in our cohort. In addition, the HR was unstable across the different cut-offs tested: The HR was close to 1–1.2 for the cut-offs between 5.5 and 6.7, and then decreased below 1 for cut-offs ranging from 6.7 to 7.2, increasing again above 2 for higher RBC counts. This low overall significance and the poor stability would suggest the “best cut-off identified” is at risk of being a type I error. Alternatively, such poor HR stability may suggest that only higher RBC counts have a prognostic significance in our cohort.

Additionally, for most tested hematological parameters, apart from RBC count and NMR, the HR was stable (i.e., HR consistently above or below 1) across the multiple cut-offs tested. Although there are no accepted criteria to define an appropriate dichotomization, a significant *p*-value only detected over only a few cut-offs and an unstable HR suggests a possible type I error [26]. In that sense, the relevance of the RBC count and NMR could be questioned here. The OSA location was also associated with PFS. Overall survival time was significantly associated with the established cut-offs for the absolute count of neutrophils, monocytes and RBC, hematocrit, and NLR and NMR. The stability and significance of tested dichotomizations for these parameters are graphically summarized in Appendix A.

On multivariable analysis (Table 3), only the cut-off associated with the absolute neutrophil count (HR 0.28 CI 95% 0.13–0.61, *p* < 0.001), RBC count (HR 3.5, CI 95% 1.56–7.9, *p* = 0.002), and OSA location (proximal humerus vs. other location, HR 3.0, CI 95% 1.48–6.1, *p* = 0.002) were significantly associated with PFS (Table 3).

The median PFS for the high- and low-neutrophil-count groups (cut-off = 4.37 × 10^9^/L) was 174 (CI 95% 140–287) and 109 days (CI 95% 100-NA), respectively (Figure 2A). The median PFS for high and low RBC count (cut-off 7.91) was 100 days (CI 95% 75-NA) and 159 days (CI 95% 133–280), respectively (Figure 2B). Dogs with OSA located in the proximal humerus location had a significantly shorter PFS (113 days, CI 95% 88-NA) than dogs with OSA at another location (174 days, CI 95% 133–277) (Figure 2C).

In the multivariable analysis, the cut-off associated with absolute neutrophil count (cut-off 4.44 × 10^9^/L, HR 0.27, CI 95% 0.13–0.53, *p* < 0.001), hematocrit (cut-off 53.05%, HR 3.38, CI 95% 1.5–7.67, *p* = 0.003), NMR (cut-off 16.8 HR 2.24, 95% CI 1.06–4.74, *p* = 0.03), and the OSA location (proximal humerus vs. other, HR 2.11, 95% CI 1.03–4.32, *p* = 0.01) were significantly associated with OST (Appendix A).

#### 3.7.2. Focus on Sighthounds

As sighthounds have different normal hematological values compared to other breeds [14] and represented 22% (13/59) of the dogs in our cohort, we assessed the relevance of the prognostic variables and their respective cut-off values detected in the multivariable analysis for the whole population, in the subpopulation of sighthounds. The distribution of the hematological parameters and associated ratios in sighthounds are highlighted in Appendix A. Given the low number of cases, only univariable analysis was performed. The cut-off values identified in the whole population for the neutrophil (*p* = 0.9), monocyte count (*p* = 0.5), and RBC count (*p* = 0.1) were not associated with PFS in sighthounds, whereas the OSA location (*p* = 0.03) was. Sighthounds with OSA located on the proximal humerus had a median PFS of 88 days (95% CI 34-NA) whereas those with OSA located elsewhere had a median PFS of 134 days (95% CI 104-NA). For OST, only the NMR (*p* = 0.003) cut-off identified in the whole population and the OSA location (*p* = 0.03) were significantly associated with outcomes in sighthounds.

This suggested that the cut-off identified for the whole population may not be applicable to sighthounds, thereby limiting the relevance of the cut-off values identified in the whole population. Therefore, we repeated the statistical analysis for a subpopulation in which sighthounds were excluded.

#### 3.7.3. Sighthounds-Excluded Population

In the univariable analysis, the best cut-off identified was significantly associated with PFS for the absolute count of neutrophils, monocytes, and RBC (Table 4). Seven out of the twenty-three (30.4%) cut-off values tested were significantly associated with PFS for the neutrophil count, whereas it was only 5.3% and 11.5% for the monocyte and RBC count, respectively (Figure 3).

The HRs were stable overall across the cut-off tested for these parameters. As before, the OSA location was associated with PFS (Table 4). Similarly, an association between OST and the absolute count of neutrophils and monocytes, as well as PNR (Appendix A)m was identified. The stability and significance of tested dichotomizations for these parameters are graphically summarized in Appendix A.

In the multivariable analysis (Table 4), only the cut-off associated with absolute neutrophil count (HR 0.29, CI 95% 0.13–0.64, *p* = 0.002), absolute monocyte count (HR 0.44, CI 95% 0.22–0.87, *p* = 0.01), and OSA location (HR 3.19, CI 95% 1.42–7.19, *p* = 0.01) were significantly associated with PFS.

The median PFS for the high- and low-neutrophil groups (cut-off = 4.44 × 10^9^/L) was 184 days (CI 95% 142–366) and 112 days (CI 95% 85-NA), respectively (Figure 4A). The median PFS for the high- and low-monocyte groups (cut-off = 0.43 × 10^9^/L) was 183 days (CI 95% 142–647) and 133 days (CI 95% 106–261), respectively (Figure 4B). The median PFS for dogs with OSA located on the proximal humerus was 128 days (CI 95% 100-NA) while it was 182 days (CI 95% 133–351) for other locations (Figure 4C). For OST, only the cut-off value of the absolute neutrophil count was associated with OST in multivariable analysis (cut-off 4.44 × 10^9^/L; HR 0.32, CI 95% 0.11–0.57, *p* = 0.006, Appendix A).

## 4. Discussion

Building on the role of hematological parameters in human OSA [15,16,17,18,19,20,21,22,31,32,33,34] and the limited information available in canine OSA [6,7,35], the primary objective of this study was to re-evaluate the role of hematological parameters in canine appendicular OSA. To do so, we retrospectively reviewed a cohort of dogs treated uniformly and applied a well-described statistical tool [26,27], investigating the prognostic value of various parameters including the platelet count, the platelet-related ratio, and SII, which had not been previously assessed in canine OSA. Our study revealed that for both PFS and OST, the neutrophil count was a prognostic marker, with dogs having a higher count showing a more favorable outcome than those with a lower count. Cut-offs with possible prognostic value for PFS were also established for other parameters such as the monocyte and the RBC counts; moreover, the negative prognostic value of proximal humerus location was confirmed [5]. We have also shown that the cut-offs established in the whole population may not be appropriate in sighthounds, suggesting that breed-specific studies are likely necessary to better understand the prognostic role of certain indicators such as hematological parameters.

In this study, we used an original statistical approach to identify possible hematological prognostic markers. Conversely to most receiver-operating-characteristic (ROC) curve analyses, the “minimal *p*-value approach”, abundantly used in human oncology [27,36], is unbiased by a pre-determined binary outcome and takes into account the time to event. This approach thus minimizes the loss of information inherent to ROC analysis [37]. The graphs showing the variation of HR for each cut-off tested allowed us to assess the stability and significance of the dichotomizations evaluated, thereby informing on the relevance of the optimal cut-off values identified (Figure 1 and Figure 3) [27].

In our population, a neutrophil count below the cut-off of 4.37 × 10^9^/L was significantly associated with shorter PFS, and this significance was retained when sighthounds were excluded from the analysis (cut-off 4.44 × 10^9^/L). The minimal difference among the cut-off values identified was most likely due to the lower neutrophil count reported in sighthounds [14]. Similarly, OST was significantly shorter for dogs with a neutrophil count <4.44 × 10^9^/L. The neutrophil count was the best dichotomization identified in our cohort, as a large proportion of the cut-offs tested was significantly associated with outcomes and the HR was stable overall over the multiple cut-offs evaluated. Interestingly, cut-offs for a low and high neutrophil count were associated with a significant survival difference (Figure 1A and Figure 3A); however, the *p*-value was lower for the low neutrophil cut-off. The favorable prognostic role of a high neutrophil count was partly unexpected, as this was not found to be associated with disease-free interval (DFI) in previous veterinary literature [6]. In human oncology, a high neutrophil count is usually associated with poorer survival [38,39,40], possibly due to their possible pro-tumorigenic effect [41,42]. However, the favorable role of the peripheral neutrophil count was recently identified in a human study on OSA [40] and the presence of neutrophils in the human OSA microenvironment has shown a protective effect against the metastatic process [43], suggesting that the role of neutrophils in OSA remain controversial. Such discrepancy in the prognostic role of neutrophils across studies is also documented in other human neoplasia such as colorectal cancer or gastric carcinoma. This might be due to the context-dependent role of the tumor-associated neutrophils (TANs) described in mouse models, in which the phenotype of TANs can vary from an N1 (anti-tumoral) to an N2 (pro-tumoral). Whereas N2 neutrophils can promote tumor progression and inhibit anti-tumor T-cell response, N1 neutrophils can have direct anti-tumor effects and T-cell stimulatory effects [44]. However, the presence of these two types of TANs and the relationship between the TAN phenotype and the peripheral neutrophil count still need to be confirmed in human cancer. In canine OSA, the improved survival reported when perioperative infection occurs after limb-sparing surgery might also corroborate the protective role of neutrophils [45,46]. Therefore, for both biological and statistical reasons, it seems likely that the neutrophil cut-off we established is appropriate to predict prognosis in canine appendicular OSA. Based on the survival difference also observed for a high neutrophil cut-off, it is possible that both a low and a high neutrophil cut-off have negative prognostic value, but our cohort was too small to analyze this further.

Exclusively in the sighthound-excluded population, the monocyte count <0.43 × 10^9^/L was associated with decreased PFS. This significance in the multivariable analysis was surprising as only 5.3% of cut-offs tested were significantly associated with PFS, raising the suspicion of type I error. This doubt on the favorable prognostic role of a high monocyte count is exacerbated by the contradiction with existing evidence. It is well known that tumor-associated macrophages (TAMs) have a pro-tumorigenic effect [47] and Sottnik *and colleagues* reported that dogs with OSA with a monocyte count >0.4 × 10^9^/L had a shorter DFI [6]. Conversely, another study [48] suggested that a high TAMs tumor infiltrate was associated with improved outcomes in canine OSA, favoring a positive prognostic role. Interestingly, whereas the study by Sottnik *and colleagues* [6] included dogs treated with carboplatin, doxorubicin, and their combination, we only included dogs treated with carboplatin similar to the study by Withers *and colleagues* [48]. As both human [49] and murine [50] studies suggest that carboplatin may enhance monocyte/macrophage function, which is impaired in canine OSA [35], it is possible that treatment may play a role in modifying the effect of TAMs and/or circulating monocytes. Further studies are thus required to elucidate the role of monocytes/macrophages in canine OSA.

The proximal humerus location was associated with shorter PFS in multivariable analysis for the whole population. This result confirms a previous meta-analysis study in which canine OSA located in the proximal humerus was associated with a poorer prognosis [5].

Interestingly, both the neutrophil and monocyte count did not play a role in the sighthound subpopulation. Concomitantly, several markers including RBC count, hematocrit, PNR, and NMR had a reverse association with PFS when comparing univariable analyses in the whole cohort and the sighthound-excluded cohort. This discrepancy might be due to statistical error but may also suggest that hematological parameters do not play a similar prognostic role in all breeds [14]. It is possible that the cut-off values obtained in the sighthound-excluded population are more generalizable than the cut-offs identified in the whole cohort. Following this hypothesis, the relevance of the RBC count as a prognostic marker is questionable, as this was only significant in the whole population; this is corroborated by the poor stability of the HR over the RBC cut-offs evaluated.

The lymphocyte count had no prognostic role, whereas a lymphocyte count >1 × 10^9^/L was associated with shorter DFI in Sottnik *and colleagues’* study [6]. A type II error is considered unlikely here based on the high sensitivity of the minimal *p*-value approach in detecting cut-off values with prognostic relevance. It also seems biologically logical that the lymphocyte count lacked a prognostic role in this study due to the possible differential contribution of the multiple lymphocyte subtypes that cannot be assessed when analyzing lymphocytes as a whole [7,51,52]. Similarly, neither the ALP levels nor the time between amputation and chemotherapy initiation were associated with outcomes in our study. The prognostic value of total serum ALP is well-recognized overall [5] but recent evidence questioned the importance of ALP as a prognostic factor. Indeed, recent evidence suggests that ALP could merely be a surrogate marker of disease burden [53] rather than a marker of aggressive biological behavior. This is supported by the similar transcriptional profiles displayed by canine OSA cell lines derived from dogs with low and high ALP [54]. Our study would then corroborate the lack of prognostic significance of ALP for dogs with similar stages, but further studies are necessary to confirm that. Alternatively, given the possible variations of total serum ALP unrelated to OSA, and the low number of dogs with ALP above the reference range, the prognostic relevance of ALP may have been missed in our cohort. Recently, it was reported that dogs starting chemotherapy within 5 days of amputation had longer disease-free intervals [25]. In our cohort, only one dog started chemotherapy ≤5 days after surgery, thereby impeding any statistical consideration.

When comparing the results of OST and PFS, several prognostic markers differed. The proximal humerus location was not associated with OST in the sighthound-excluded population, while a cut-off was identified for NMR or PNR only when OST was the outcome of interest. The discrepancy between PFS and OST may be due to the subjectivity in the decision for euthanasia and may be exacerbated by the variability of the rescue treatments used at the time of progression. Therefore, the hematological prognostic markers identified for OST alone should be considered with caution, and further studies with larger cohorts will be required to confirm their relevance.

Our study has many limitations associated with its retrospective nature, the stringent inclusion criteria, and therefore the low number of cases, possibly contributing to a type II error. Additionally, different analyzers were used, and previous studies have suggested a possible variation of 20–30% between instruments, at an individual level [55]. However, this study also suggested that, at a cohort level, the mean difference between in-office instruments and the ADVIA 120 was zero, suggesting that, when populations are analyzed as a whole, the white blood cell count is overall similar between analyzers. Due to the use of several analyzers, we also made the choice of using an international reference range, keeping in mind the limitations of this approach [56]. This means that the detection of neutropenia or thrombocytopenia may be inaccurate but will not have any impact on the assessment of the cut-off identified, established independently from any reference range. Moreover, the frequency and modality of restaging were not homogeneous, and progression was not cytologically/histopathologically confirmed in all cases, thereby limiting the accuracy of the date of progression. Therefore, PFS was preferred compared to other survival endpoints such as the disease-free interval. Furthermore, although the median time between hematology and amputation was only six days, we included dogs for which hematology was performed up to 21 days prior to amputation, and hematological parameters may have changed during this time. There is no standard to define the “pre-operative” period, and this time varies across veterinary and human studies, sometimes reaching up to 60 days in veterinary [9,12] and 30 days in human studies [57,58]. Furthermore, although most dogs were staged thoroughly, some dogs might have suffered from unidentified diseases, causing alterations in their hematological parameters. Finally, given the risk of false discovery of the statistical method elected and the small size of our population, the use of a validation cohort would be indispensable to confirm our findings [27].

## 5. Conclusions

Our study suggests that pre-treatment inflammatory indices, primarily including the absolute neutrophil and, possibly, monocyte counts, may play a prognostic role in canine OSA treated with amputation and adjuvant carboplatin. Our study also suggests that hematological markers identified in the whole population may not be applicable to sighthounds. These pre-treatment indices, which could prove very helpful as they are readily available, should be confirmed in large prospective studies taking into account breed specificities.

## Figures and Tables

**Figure 1 vetsci-10-00495-f001:**
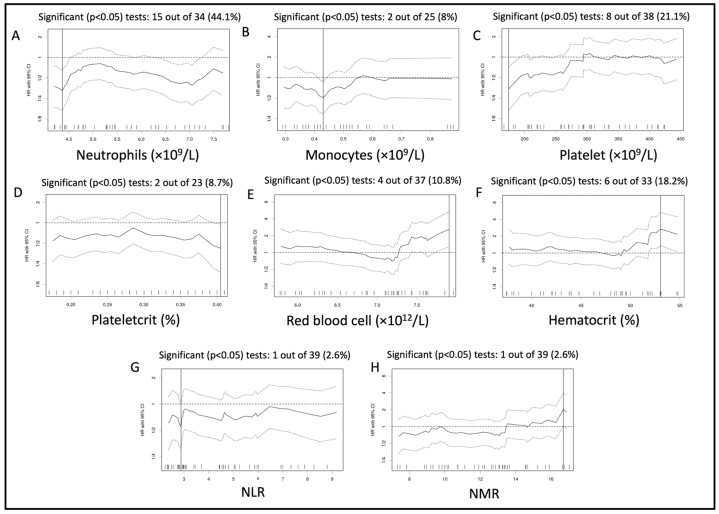
Variation of hazard ratio per cut-off assessed for the neutrophil count (**A**), monocyte count (**B**), platelet count (**C**), plateletcrit (**D**), red blood cell (RBC) count (**E**), hematocrit (**F**), NLR (**G**), and NMR (**H**) analyzed for progression-free survival of the whole population. Solid lines represent the hazard ratio (HR), and the dotted line represents the 95% confidence interval around each HR (95% CI).

**Figure 2 vetsci-10-00495-f002:**
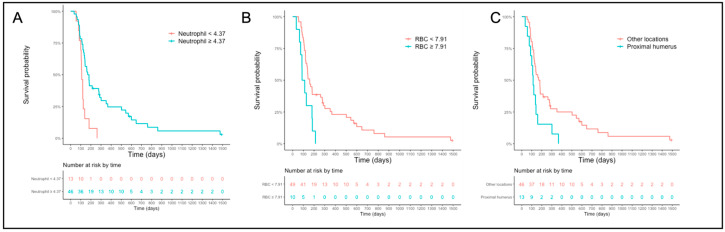
Progression-free survival curves for the whole population based on the cut-off established for the neutrophil count (**A**), red blood cell (**B**), and primary tumor location (**C**). Abbreviation: RBC = red blood cells.

**Figure 3 vetsci-10-00495-f003:**
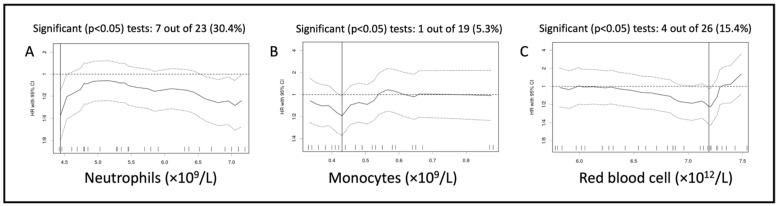
Variation of hazard ratio per cut-off assessed for the neutrophil count (**A**), monocyte count (**B**), and red blood cell count (**C**) analyzed for progression-free survival of the sighthound-excluded population. Solid lines represent the hazard ratios (HR), and the dotted line represents the 95% confidence interval around each HR (95% CI).

**Figure 4 vetsci-10-00495-f004:**
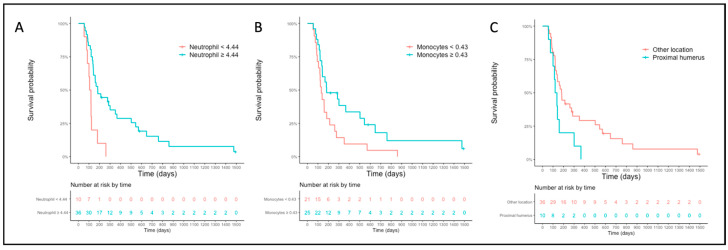
Progression-free survival curves for the sighthound-excluded populations based on the cut-off established for the neutrophil count (**A**), monocyte count (**B**), and primary tumor location (**C**).

**Table 1 vetsci-10-00495-t001:** Reference range of hematological and biochemical parameters used. This table presents the internationally accepted reference range that was used in our study based on Schalm’s Veterinary Hematology [24].

Parameters	Reference Interval
Red blood cells	5.5–8.5 × 10^12^/L
Hematocrit	0.37–0.55 L/L
Plateletcrit *	0.1–0.5%
Neutrophils	3–11.5 × 10^9^/L
Monocytes	0.15–1.35 × 10^9^/L
Lymphocytes	1–4.8 × 10^9^/L
Platelet count	200–500 × 10^9^/L
Mean Platelet volume	6.7–11.1 fL
Alkaline phosphatase	0–140 U/L

* the plateletcrit reference range was calculated based on the MPV and platelet count reference range obtained from Schalm’s Veterinary Hematology [24].

**Table 2 vetsci-10-00495-t002:** Patient demographics of the whole population of 59 dogs.

Parameters	Descriptive Statistics	Reported Numbers
Breed	Number (percentage)	- Crossbreeds 9 (15.25%) - Rottweiler 9 (15.25%) - Greyhound 9 (15.25%) - German Shepherd 7 (11.86%) - Labrador 6 (10.17%)
Age (years)	Median (range)	8.8 (1.5–13.5)
Body weight (kg)	Median (range)	33 (6.2–54)
Sex	Number (percentage)	- FN 26 (44.1%) - FE 8 (13.56%) - MN 11 (18.64%) - ME 14 (23.7%)
Tumor location	Number (percentage)	- Femur 15 (25.42%) - Tibia 15 (25.42%) - Proximal humerus 13 (22.03%) - Radius 12 (20.34%) - Ulna 2 (3.39%) - Scapula 1 (1.7%) - Mid-humerus 1 (1.7%)
Thoracic imaging	Number (percentage)	- CT 47 (79.7%) - XR 12 (20.3%)

Abbreviations. CT: computed tomography, FE: female entire, FN: female neutered, ME: male entire, MN: male neutered, ME: male entire, XR: radiographs.

**Table 3 vetsci-10-00495-t003:** Univariable and multivariable analysis—association of progression-free survival with possible prognostic factors in the whole population.

		Univariable Analysis		Multivariable Analysis	
	Cut-Off	HR (95% CI)	*p*-Value	HR (95% CI)	*p*-Value
**N**	4.37	0.33	**<0.001**	0.28	**0.001**
		(0.16–0.64)		(0.13–0.61)	
**M**	0.43	0.51	**0.013**	0.55	0.06
		(0.29–0.87)		(0.3–1.02)	
**PLT**	174.5	0.34	**0.003**	-	-
		(0.16–0.71)			
**MPV**	11.25	0.61	0.19	-	-
		(0.28–1.29)			
**PCT**	0.41	0.42	**0.03**	-	-
		(0.19–0.95)			
**L**	1.75	1.55	0.14	-	-
		(0.86–2.81)			
**RBC**	7.91	2.6	**0.007**	3.5 (1.56–7.9)	**0.002**
		(1.26–5.33)			
**HCT**	53.05	2.68	**0.003**	-	**-**
		(1.35–5.3)			
**NLR**	2.88	0.55	**0.04**	-	-
		(0.3–0.99)			
**LMR**	1.87	1.45	0.23	-	-
		(0.79–2.67)			
**PLR**	158.9	0.58	0.073	-	-
		(0.32–1.06)			
**PNR**	33.08	0.51	0.06	-	**-**
		(0.25–1.03)			
**PMR**	1065	1.46	0.23	-	-
		(0.78–2.72)			
**NMR**	16.7	2.01	**0.04**	-	-
		(1.02–3.95)			
**SII**	1243	0.63	0.09	-	-
		(0.37–1.08)			
**Sex**		0.93	0.8	-	-
**(Male vs. Female)**		(0.54–1.6)			
**AGE**	7.5	0.73	0.29	-	-
		(0.41–1.31)			
**BW**	39.8	0.65	0.19	-	-
		(0.34–1.24)			
**Staging modality**	-	0.75	0.4	-	-
**(CT vs. XR)**		(0.38–1.47)			
**ALP**	133.5	1.4	0.25	-	-
		(0.79–2.51)			
**OSA location**	-	2.25 (1.18–4.3)	**0.01**	3.0 (1.48–6.1)	**0.002**
**(Prox. hum vs. other)**					
**Amputation to chemotherapy**	14.5	0.67 (0.34–1.31)	0.24		

Abbreviations. 95% CI: 95% confidence interval, ALP: Alkaline phosphatase, BW: Body weight, CT: Computed tomography, HCT: Hematocrit, HR: Hazard ratio, L: Lymphocyte count, LMR: Lymphocyte/monocyte ratio, M: Monocyte count, MPV: Mean platelet volume, N: Neutrophil count, NLR: Neutrophil/lymphocyte ratio, NMR: Neutrophil/monocyte ratio, PCT: Plateletcrit, PLR: Platelet/lymphocyte ratio, PLT: Platelet count, PMR: Platelet/monocyte ratio, PNR: Platelet/neutrophil ratio, Prox. Hum.: Proximal humerus, RBC: Red blood cell count, SII (systemic inflammatory index): [(PLT × N)/L], XR: Radiographs. Significant *p*-values in bold.

**Table 4 vetsci-10-00495-t004:** Univariable and multivariable analysis—association of progression-free survival with possible prognostic factors in the population after the exclusion of sighthounds.

		Univariable Analysis		Multivariable Analysis	
	Cut-Off	HR (95% CI)	*p*-Value	HR (95% CI)	*p*-Value
**N**	4.44	0.27	**<0.001**	0.29	**0.002**
		(0.12–0.6)		(0.13–0.64)	
**M**	0.43	0.51	**0.03**	0.44 (0.22–0.87)	**0.01**
		(0.28–0.95)			
**PLT**	305	1.47	0.22	-	-
		(0.79–2.75)			
**MPV**	8.45	0.7	0.28	-	-
		(0.36–1.34)			
**PCT**	0.405	0.48	0.08	-	-
		(0.21–1.1)			
**L**	1.745	1.71	0.11	-	-
		(0.88–3.32)			
**RBC**	7.19	0.46	**0.02**	-	-
		(0.23–0.92)			
**HCT**	49.05	0.52	0.08	-	-
		(0.25–1.11)			
**NLR**	2.875	0.53 (0.27–1.02)	0.053	-	-
**LMR**	1.87	1.58	0.22	-	-
		(0.76–3.32)			
**PLR**	158.9	0.68	0.28	-	-
		(0.34–1.38)			
**PNR**	77.16	1.97	0.055	-	-
		(0.97–3.98)			
**PMR**	1065	1.53	0.22	-	-
		(0.77–3.06)			
**NMR**	12.8	0.58	0.1	-	-
		(0.3–1.11)			
**SII**	1243	0.68	0.22	-	-
		(0.37–1.26)			
**Sex**		1.05	0.80	-	-
**(Male vs. Female)**		(0.56–1.93)			
**AGE**	8.25	0.73	0.33	-	-
		(0.38–1.39)			
**BW**	25.65	1.56	0.22	-	-
		(0.77–3.17)			
**Staging modality**	-	0.68	0.36	-	-
**(CT vs. XR)**		(0.30–1.55)			
**ALP**	133.5	1.42	0.3	-	-
		(0.73–2.75)			
**OSA location**	-	2.14 (1.02–4.5)	**0.04**	3.19 (1.42–7.19)	**0.01**
**(Prox. hum vs. other)**					
**Amputation to chemotherapy**	14.5	0.61 (0.3–1.28)	0.19		

Abbreviations: 95% CI: 95% confidence interval, ALP: Alkaline phosphatase, BW: Body weight, CT: Computed tomography, HCT: Hematocrit, HR: Hazard ratio, L: Lymphocyte count, LMR: Lymphocyte/monocyte ratio, M: Monocyte count, MPV: Mean platelet volume, N: Neutrophil count, NLR: Neutrophil/lymphocyte ratio, NMR: Neutrophil/monocyte ratio, PCT: Plateletcrit, PLR: Platelet/lymphocyte ratio, PLT: Platelet count, PMR: Platelet/monocyte ratio, PNR: Platelet/neutrophil ratio, Prox. Hum.: Proximal humerus, RBC: Red blood cell count, SII (systemic inflammatory index): [(PLT × N)/L], XR: Radiographs. Significant *p*-values in bold.

## Data Availability

The data that support the findings of this study are available from the corresponding author upon reasonable request.

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
