# Peer review of "The Prognostic Role of Preoperative Hematological and Inflammatory Indices in Canine Appendicular Osteosarcoma"

_vetsci, 2023, doi:10.3390/vetsci10080495_

Round 1
Reviewer 1 Report
The present study analyzes routinely assessed blood cell counts in dogs suffering of appendicular osteosarcoma. This multicenter study allowed them to include a relatively large, well-defined and homogeneous cohort of different breeds, subjected to the same treatment regimen. Blood parameters are routinely assessed before surgery and thus can be cheap and easy prognostic parameters, if related to outcome. The methodology of iteratively approaching cut-off values with the lowest p-value and the obtained results are well-explained and clearly presented. After excluding sighthounds, which have different reference intervals compared to other breeds, the authors identified a low neutrophil count, a low monocyte count and a proximal humerus location of OSA as negative prognostic values. The study is well-designed and the results are discussed critically.
I have a few minor comments and questions, mostly regarding the methodology:
- Line 45: As a suggestion, you could rearrange the sentence for a better reading flow and start with the sighthound-only population.
- Line 146: I have problems with the word „censored“. Did you exclude the patients lost to follow-up from the PFS and OST, or were the last appointments set as relapse or death? The same question comes up in lines 256-259: I cannot distinguish between patients, which died without previously documented progression and dogs lost to follow-up in Table S1.
- Lines 341, 367 and 368: This is maybe connected to my last comment, I cannot find patients with NA PFS and OST in Table S1.
- Line 154: Maybe you could explain HR as hazard ratio of relapse or death.
- Table 1: Some parameters are without units.
- Line 107 and 263: It is a bit confusing, did you exclude patients which received chemotherapy in addition to carboplatin as initial treatment and allowed chemo as rescue treatment? Could you add that to line 107?
- Lines 358 and 518: Here is something wrong with the format of the references.
Reviewer 2 Report
The paper deals with a relevant and challenging topic and represents an important and applicable contribution to cancer pacients. The text is well constructed, the methodology is clearly described and the bibliography updated. In order to contribute to the understanding of the pathophysiology of the process, it is suggested to include the parameters histopathological graduation and tumor stage in the analysis. Finally, it would be desirable to expand the discussion on neutrophilic activity in oncologic disease.
